# 1-Carbomethoxy-β-Carboline, Derived from *Portulaca oleracea* L., Ameliorates LPS-Mediated Inflammatory Response Associated with MAPK Signaling and Nuclear Translocation of NF-κB

**DOI:** 10.3390/molecules24224042

**Published:** 2019-11-07

**Authors:** Kang-Hoon Kim, Eun-Jae Park, Hyun-Jae Jang, Seung-Jae Lee, Chan Sun Park, Bong-Sik Yun, Seung Woong Lee, Mun-Chual Rho

**Affiliations:** 1Immunoregulatory Material Research Center, Korea Research Institute of Biotechnology, 181 Ipsin-gil, Jeongeup-si, Jeonbuk 56212, Koreapej911029@kribb.re.kr (E.-J.P.); water815@kribb.re.kr (H.-J.J.); seung99@kribb.re.kr (S.-J.L.); chansun@kribb.re.kr (C.S.P.); 2Division of Biotechnology and Advanced Institute of Environment and Bioscience, College of Environmental and Bioresource Sciences, Chonbuk National University, Iksan-si 570-752, Korea; bsyun@jbnu.ac.kr

**Keywords:** *Portulaca oleracea*, 1-carbomethoxy-β-carboline, anti-inflammation, MAPKs, NF-κB

## Abstract

*Portulaca oleracea* is as a medicinal plant known for its neuroprotective, hepatoprotective, antidiabetic, antioxidant, anticancer, antimicrobial, antiulcerogenic, and anti-inflammatory activities. However, the specific active compounds responsible for the individual pharmacological effects of *P. oleracea* extract (95% EtOH) remain unknown. Here, we hypothesized that alkaloids, the most abundant constituents in *P. oleracea* extract, are responsible for its anti-inflammatory activity. We investigated the phytochemical substituents (compounds **1–22**) using nuclear magnetic resonance (NMR) and electrospray ionization mass spectrometry (ESI-MS) and screened their effects on NO production in lipopolysaccharide (LPS)-induced macrophages. Compound **20**, 1-carbomethoxy-β-carboline, as an alkaloid structure, ameliorated nitric oxide (NO) production, inducible nitric oxide synthase (iNOS), and proinflammatory cytokines associated with the mitogen-activated protein kinase (MAPK) pathways, p38, extracellular signal-regulated kinase (ERK), and c-Jun N-terminal kinase (JNK). Subsequently, we observed that compound **20** suppressed nuclear translocation of nuclear factor κB (NF-κB) using immunocytochemistry. Moreover, we recently reported that compound **8**, *trans*-*N*-feruloyl-3’, 7’-dimethoxytyramine, was originally purified from *P. oleracea* extracts. Our results suggest that 1-carbomethoxy-β-carboline, the most effective anti-inflammatory agent among alkaloids in the 95% EtOH extract of *P. oleracea,* was suppressing the MAPK pathway and nuclear translocation of NF-κB. Therefore, *P. oleracea* extracts and specifically 1-carbomethoxy-β-carboline may be novel therapeutic candidates for the treatment of inflammatory diseases associated with the activation of MAPKs and NF-κB.

## 1. Introduction

*Portulaca oleracea* L. (*P*. *oleracea*), an annual succulent belonging to the family Portulacaceae, may grow to a height of 40 cm in many parts of the Mediterranean and tropical Asian countries [1]. Although *P*. *oleracea* tastes salty and sour, it has been consumed for its nutrients [2] and as a traditional medicine as a febrifuge, antiseptic, and vermifuge [3]. Moreover, *P*. *oleracea* is safe as it has been prescribed for thousands of years as a traditional Chinese medicine as well as in recent clinical studies [4], and the World Health Organization (WHO) listed *P*. *oleracea* as being effective as a global panacea in 1990. Recent studies have shown that *P*. *oleracea* extracts have diverse pharmacological effects including neuroprotective, hepatoprotective, antidiabetic, antioxidant, anticancer, antimicrobial, antiulcerogenic, and anti-inflammatory activities with minimal side effects [5]. The diverse effects seem to be caused by the presence of many phytochemical constituents in *P*. *oleracea* extracts, such as vegetable sources of omega-3 fatty acids, polyphenols, flavonoids, terpenoids, fatty acids, ferulic amide, and alkaloids [6,7].

Among these phytochemicals in *P*. *oleracea* extracts, the most abundant are alkaloids such as aurantiamideacetate, aurantiamide, 1,5-dimethyl-6-phenyl-1,6,3,4-tetrahydro-1,2,4-2(1H)-triazine, trollisine, cyclo(_L_-tyrosinyl-_L_-tyrosinyl), 3,5-bis(3-methoxy,4-hydroxyphenyl)-5,6-dihydro,2(1H)-pyridinone, N-feruloyl normetanephrine, N-trans-feruloyl tyramine, and 1-carbomethoxy-beta-carboline [8,9,10,11]. In terms of bioactivity, alkaloids derived from medicinal plants such as *Hydrastis canadensis*, *Coptidis rhizome*, *Phellodendri cortex*, and *Berberis crataegina* have shown novel anti-inflammatory responses involving the suppression of the nuclear translocation of nuclear factor κB (NF-κB) and the inhibition of nitric oxide (NO) production or proinflammatory cytokines such as tumor necrosis factor alpha (TNF-α), interleukin-6 (IL-6), and interleukin-1β (IL-1β) [12].

NF-κB acts on a family of inducible transcription factors that regulate a large array of genes engaged in different processes in immune response and inflammation in innate immune cells, including macrophages [13]. The canonical NF-κB pathway responds to several stimuli, including ligands of various cytokine receptors, pattern-recognition receptors (PRRs) [14]. In particular, mammalian cells express five families of PRRs, namely, Toll-like receptors (TLRs), RIG-I-like receptors, nucleotide-binding and oligomerization domain (NOD)-like receptors (NLRs), C-type lectin-like receptors, and cytosolic DNA sensors [15]. Among these PRRs, toll-like receptor 4 (TLR4) ligand lipopolysaccharide (LPS) leads to macrophage differentiation toward the M1 phenotype [16]. NF-κB is a novel transcription factor of M1 macrophages and is required for the induction of diverse inflammatory genes, including those encoding TNF-α, IL-6, and IL-1β [17]. Additionally, mitogen-activated protein kinase (MAPK) pathways, associated with extracellular signal-regulated kinase (ERK), c-Jun N-terminal kinase (JNK), and p38 kinase, enhance NO production and inducible inducible nitric oxide synthase (iNOS) by the activation of the nuclear translocation of NF-κB [18]. If a natural compound suppresses MAPK pathways and reduces the activation of the nuclear translocation of NF-κB, it may be a potential therapeutic agent for the treatment of inflammatory diseases.

Although *P*. *oleracea* extracts exhibit diverse pharmacological effects, including anti-inflammatory activity, individual components of *P*. *oleracea* extracts must be separated and the bioactivity of each component should be evaluated prior to the therapeutic application of this material in clinical trials or as a health functional food. In our study, we first elucidated the chemical structures of the phytochemical constituents (compounds **1** to **22**) from *P*. *oleracea* extracts (95% EtOH) using spectroscopic data, including NMR and ESI-MS. Among these constituents, this is the first report that 1-carbomethoxy-β-carboline, an alkaloid structure, significantly ameliorated NO production and proinflammatory cytokines associated with the MAPK pathway and the nuclear translocation of NF-κB under LPS-mediated inflammatory conditions in RAW 264.7. Additionally, we originally report the identification of a new compound, *trans*-*N*-feruloyl-3’, 7’-dimethoxytyramine, from the *P*. *oleracea* extract.

## 2. Results and Discussion

### 2.1. Isolation and Structural Elucidation of Compounds

We used a variety of separate purification procedures to find components that exhibit anti-inflammatory activity from *P*. *oleracea* extracts (95% EtOH). Twenty-two compounds (Figure 1), namely, a new ferulic amide (**8**) and 21 known compounds, including 10 ferulic amides (**1**–**7** and **9**–**11**) and 11 phenolic compounds (**12**–**22**), were isolated from the *P*. *oleracea* extracts. The structures of the isolated compounds were identified by spectroscopic data acquired using ^1^H NMR, ^13^C NMR, ^1^H-^1^H correlated spectroscopy (COSY), heteronuclear multiple bond correlation (HMBC), heteronuclear multiple quantum correlation (HMQC), high resolution electrospray ionization mass spectrometry (HRESIMS), infrared (IR) spectroscopy, and polarimetry techniques. By comparing the NMR data of the isolated compounds with data reported in the literature, the known compounds were identified as *trans-N*-courmaroyltyramine (**1**) [19], *trans-N-*feruloyloctopamine (**2**) [20], *trans-N-*feruloyltyramine (**3**) [19], (7’*S*)-*trans*-*N*-feruloyloctopamine (**4**) [21], (*S*)-3-(4-hydroxy-3-methoxyphenyl)-*N*-(2-(4-hydroxyphenyl)-2-methoxyethyl)-acrylamide (**5**) [22], *trans*-*N*-feruloyl-3’-methoxytyramine (**6**) [21], (7’S)-*trans*-*N*-feruloylnormetanephrine (**7**) [21], *N*-*trans*-hibiscusamide (**9**) [21], perillascens (**10**) [23], aurantiamide acetate (**11**) [24], methyl *N*-acetylphenylalaninate (**12**) [25], *N*-benzoyl-l-aspartic acid dimethyl ester (**13**) [26], *N*-benzoyl-d-aspartic acid dimethyl ester (**14**) [26], scinamide C (**15**) [27], oleracein E (**16**) [28], methyl hematinate (**17**) [29], 3-formylindole (**18**) [30], 3-propionylindole (**19**) [30], 1-carbomethoxy-beta-carboline (**20**) [11], methyl pyroglutamate (**21**) [31], and ethyl pyroglutamate (**22**) [32].

Additionally, we newly isolated compound **8**, as a yellowish oil (Figure 2). A molecular formula of C_20_H_23_NO_6_ was determined based on its HRESIMS spectrum and was deduced from ion peak at *m/z* 396.1423 [M + Na] ^+^ (396.1423, calcd for C_20_H_23_O_6_Na, Appendix A). The ultraviolet (UV) spectrum showed absorption peaks at 200, 227, 283, and 320 nm. The IR spectrum suggested the presence of OH groups, as indicated by a wide band at 3356 cm^−1^ and an amide carbonyl, as indicated by bands at 1737 cm^−1^ and 1653 cm^−1^ (Appendix A). The ^1^H NMR spectrum exhibited signals for three methoxy groups at δ_H_ 3.89 (3H, s, OCH_3_-3), 3.86 (3H, s, OCH_3_-3’), and 3.24 (3H, s, OCH_3_-7’); two olefinic protons at δ_H_ 7.44 (1H, d, *J* = 15.6 Hz, H-7) and 6.45 (1H, d, *J* = 15.6 Hz, H-8); two ABX aromatic ring protons at δ_H_ 7.13 (1H, d, *J* = 1.8 Hz, H-2), 7.03 (1H, dd, *J* = 8.4, 1.8 Hz, H-6), 6.92 (1H, s), and 6.80 (3H, overlap); one methylene proton at δ_H_ 3.53 (1H, dd, *J* = 13.8, 4.8 Hz, H-8’α) and 3.42 (1H, dd, *J* = 13.8, 8.4 Hz, H-8’β); and one methine proton at δ_H_ 4.26 (1H, dd, *J* = 8.4, 4.8 Hz, H-7’) (Appendix A). The ^13^C NMR spectrum showed 20 resonances, which were classified as three methoxy carbons, *sp*^2^ carbon signals corresponding to two benzene rings, one carbonyl, one methylene, one methane, and one alkyne (Appendix A). These proton and carbon assignments were further confirmed by detailed analysis of the ^1^H-^1^H COSY, HMQC, and HMBC spectra. The ^1^H-^1^H COSY spectrum showed correlations of H-7/H-8, H-7’/H-8’, and H-6/H-5 (Appendix A). The HMQC spectrum exhibited correlations from the aromatic ring proton at 6.80 ppm to C-5, C-5’, and C-6’, indicating that the aromatic ring protons overlap with three of the aromatic ring carbons (Appendix A). Additionally, the HMBC spectrum showed cross-peaks for 3-OCH_3_/C-3, 3’-OCH_3_/C-3’, and 7’-OCH_3_/C-7’, indicating the positions of the methoxy groups. The HMBC cross-peaks of H-7/C-8, H-7/C-1, H-7/C-2, H-7/C-6, and H-6/C-7 suggested that C-7 of the alkyne was connected to the A ring. The HMBC cross-peaks of H-7’/C-2’, H-7’/C-8’, H-7’/C-1’, H-7’/C-6’, and H-2’/C-7’ suggested that C-7’ of the methine group was connected to the B ring and a methylene group (Appendix A). Except for the additional methoxy group at C-7’, the NMR signals of **8** were similar to those of compound **6** (*trans*-*N*-feruloyl-3’-methoxytyramine). Compound **8** was finally identified as *trans*-*N*-feruloyl-3’, 7’-dimethoxytyramine (Table 1, Appendix A).

### 2.2. Nitric Oxide Production Screening Using Twenty-Two Compounds Isolated from *Portulaca oleracea*

NO is a major proinflammatory mediator, involved in the pathogenesis of inflammation [33]. During an inflammatory response, the large amount of NO formed by the action of iNOS surpasses the standard physiological amount of NO [34]. A previous study reported that *P*. *oleracea* extracts had anti-inflammatory effects and ameliorated inflammatory bowel disease under dextran sulfate sodium-induced colitis by inhibiting NF-κB and MAPK activation [35,36]. However, the specific active compound responsible for the inflammatory effects remained unclear. Then, we evaluated NO production in LPS-induced RAW 264.7, a mouse macrophage cell line, upon treatment with the 22 compounds isolated from the *P*. *oleracea* extract (Figure 3).

After screening the impacts of these compounds on NO production, we selected the most potent inhibitor of NO production based on an inhibitory effect of 50% or more as the cut-off. Phenylbutazone (100 μg/mL), one of the nonsteroidal anti-inflammatory drugs, had the best inhibitory effect on nitric oxide of about 50% in RAW 264.7 cells [37]. Of these isolated compounds, compounds that inhibited NO production beyond 50% cut-off under LPS-induced inflammatory conditions were compounds **15** (scinamide C) and **20** (1-carbomethoxy-β-carboline) (Figure 3A), and their inhibitory rates were 50.92 ± 0.95% (compound **15**) and 53.33 ± 2.49% (compound **20**).

In NO production screening in Figure 3, compounds **3**, **5**, **6**, **9**, and **16** showed inhibitory effects of NO production similar to compounds **15** and **20**. However, compounds **3**, **5**, **6**, **9**, and **16** were already reported in previous studies associated with anti-inflammatory effects or cellular toxicity. Compound **3**, derived from *Arcangelisia gusanlung* and Wolfberry, and compound **6**, derived from Wolfberry, were previously reported to inhibit NO production [38,39]. Compound **5**, derived from *Solanum nigrum*, had an anti-inflammatory effect by inhibition of leukotriene C_4_ (LTC_4_) [40] and compound **16**, derived from *P*. *oleracea*, had a neuroprotective effect by reducing reactive oxygen species (ROS) and inhibiting ERK 1/2 phosphorylation [41]. Compound 9, derived from *Hibiscus tiliaceus*, had a cytotoxic effect [42]. Thus, we decided to further evaluate compounds **15** and **20**. Additionally, to investigate the cellular toxicology after treatment with compounds **15** and **20**, we tested cell viability upon treatment with the test compounds at 1 to 100 μM using the MTT assay (Figure 3B). Neither compound showed toxicological effects below 100 μM. These results suggested that the major constituents of *P*. *oleracea* extract responsible for its ability to inhibit NO production were compounds **15** and **20**.

### 2.3. Inhibitory Activities of Proinflammatory Mediators by Compounds **15** and **20**

iNOS, which is involved in the production of NO, is a novel signaling molecule associated with the MAPK pathway and NF-κB activity in both microglia and macrophage cells [43]. Subsequently, proinflammatory cytokines such as TNF-α, IL-6, and IL-1β are induced by activating nuclear translocation of NF-κB [44]. To investigate intracellular biological evidence of reduced NO level by compound **15** or **20**, we evaluated the effect of compounds **15** and **20** on the proinflammatory mediators such as iNOS, TNF-α, IL-6, and IL-1β in LPS-induced macrophages (Figure 4). In our results, compound **20** was found to be the most effective inhibitor of iNOS under LPS-induced macrophages based on Western blotting compared with compound **15** (Figure 4A). The inhibitory effect of compound **20** (12.5 µM) was similar to that of dexamethasone (positive control, 10 µM). Moreover, we tested proinflammatory cytokines after treatment with compound **15** or **20**. Treatment with compound **15** (25 µM) or **20** (12.5 µM) significantly inhibited the proinflammatory cytokine mRNA of TNF-α, IL-6, and IL-1β. There were significantly inhibitory effects of proinflammatory cytokine mRNA of TNF-α, IL-6, and IL-1β after treatment of compound **15** (25 µM) or **20** (12.5 µM). These results suggested that of the alkaloids in *P. oleracea* extracts (95% EtOH), the major active compound responsible for its anti-inflammatory activity was compound **20** (1-carbomethoxy-beta-carboline).

### 2.4. *Portulaca oleracea* Inhibits Proinflammatory and Inflammatory Signaling

If compound **20** decreased proinflammatory mediators associated with MAPKs and nuclear translocation of NF-κB with a potency similar to dexamethasone and nonsteroidal anti-inflammatory drugs (NSAIDs) [45], compound **20** may be a useful therapeutic agent for treating inflammatory diseases with minimal toxic side effects. Then, to further investigate the anti-inflammatory effects associated with the inhibition of NO production and iNOS, we evaluated the major inflammatory signaling pathways, MAPKs, associated with p38, ERK, and JNK and nuclear translocation of NF-κB in LPS-induced murine macrophages, RAW 264.7 cells upon treatment with or without compound 20. Western blotting indicated that pretreatment with compound **20** remarkably disturbed the MAPK signaling pathways associated with JNK, ERK, and p38 under LPS-induced inflammatory conditions in RAW 264.7 cells (Figure 5A). The protein expression ratios of p38, ERK, and JNK are shown in Figure 5B–D.

Subsequently, we tested prevention by pretreatment with compound **20** and qualitatively and quantitatively analyzed NF-κB (Figure 6). First, we determined whether compound **20** suppressed the nuclear translocation of NF-κB. Immunocytochemistry clearly demonstrated that compound **20** reduced the translocation of NF-κB into the nucleus in RAW 264.7 cells (Figure 6A). Furthermore, we evaluated nuclear NF-κB and cytosolic IκBα using Western blotting (Figure 6B). Compound **20** remarkably ameliorated translocation of nuclear NF-κB by suppressing the degradation of IκBα, similar to dexamethasone (Figure 6C,D). A schematic of the mechanism of compound **20** is shown in Figure 6E. These results suggest that compound **20** significantly inhibited the MAPK pathway and suppressed the degradation of IκBα along with the nuclear localization of NF-κB under LPS-induced inflammation in RAW 264.7 cells.

## 3. Materials and Methods

### 3.1. General Experimental Procedures

The structures of isolated compounds were identified by spectroscopic data including ^1^H NMR, ^13^C NMR, COSY, HMBC, HMQC, HRESIMS, IR, and optical rotation. ^1^H, ^13^C, and 2D NMR spectra were recorded on a JNM-ECA600 (Jeol, Tokyo, Japan) instrument using TMS as references. High-resolution electrospray ionization mass spectrometry (HRESIMS) was carried out using a Waters SYNAPT G2-Si HDMS spectrometer (Waters, Milford, MA, USA). IR spectra were obtained on a FT/IR-4600 (Jasco, Tokyo, Japan) spectrometer. UV spectra were recorded on a spectraMax M_2_^ϴ^ (Molecular Devices, Sunnyvale, CA, USA) spectrophotometer. Optical rotations were measured on a Jasco P-2000 polarimeter (Jasco). The HPLC analysis was performed using an Agilent 1220 infinity HPLC system (Agilent Technologies, CA, USA) equipped with a quaternary pump.

### 3.2. Plant Material

Dried *P. oleracea* (20 kg) was purchased from the Kyung-dong market in Seoul, Korea in May 2013. One of the authors (M.–C.R.) performed botanical identification, and a voucher specimen (KRIB-KR2013-003) was deposited at the laboratory of the Immunoregulatory Materials Research Center, Jeonbuk Branch of the Korea Research Institute of Bioscience and Biotechnology.

### 3.3. Extraction and Isolation

Dried powder of *P*. *oleracea* (20 kg) was refluxed with 95% EtOH (5 h, 3 × 35 L). After removal of solution under reduced pressure, the 95% EtOH extract (1.5 kg) was suspended in H_2_O and partitioned with *n*-hexane (3 × 10 L), ethyl acetate (3 × 10 L), and *n*-butanol (3 × 10 L). Each fraction was evaporated under reduced pressure. The hexane fraction (200 g) was subjected to column chromatography (CC) on silica gel (40 × 10 cm, 200 ~ 400 mesh, Merck, Kenilworth, NJ, USA) with a hexane-ethyl acetate gradient solvent system (each 1L, 1:0—0:1, *v/v*) to yield 16 fractions (POH–POH 16). POH 1, 5, 6, 8, 9, 10, and 11 were applied to silica gel CC again with a hexane-ethyl acetate gradient solvent system (each 1L, 1:0—0:1, *v/v*) and subfractions were obtained.

Compound **11** (t_R_ = 45 min, 6.2 mg) was purified from POH 6G (65 mg) by semipreparative HPLC (45% MeCN). The POH 8I (901 mg) was subjected to chromatography on a C_18_ MPLC using H_2_O and MeOH (8:2—0:10, *v/v*) to obtain 14 subfractions (POH 8I1–8I14). Compounds **18** (t_R_ 48 min, 14.6 mg) and **19** (t_R_ = 59 min, 1.9 mg) were purified from POH 8I1 (87 mg) by semipreparative HPLC (25% MeCN. The POH 10B (812 mg) was subjected to chromatography on a C_18_ MPLC using H_2_O and MeOH (2:8—0:10, *v/v*) to yield 12 subfractions (POH 10B1–POH 10B12), and POH 10B3 (132 mg) was further purified by semipreparative HPLC (25% MeCN) to obtain compound **6** (t_R_ = 32 min, 7.4 mg). POH 10C (954 mg) was chromatographed on a C_18_ MPLC using H_2_O and MeOH (2:8—0:10, *v/v*) to yield 12 subfractions (POH 10C1–POH 10C12), and POH 10C1 (103 mg) was purified by semipreparative HPLC (25% MeCN) to yield compound **3** (t_R_ = 65 min, 31 mg). Compound **1** (t_R_ = 38 min, 2.1 mg) was purified from POH 10E (403 mg) by semipreparative HPLC (20% MeCN). POH 11E (245 mg) was subjected to chromatography on a C_18_ MPLC using H_2_O and MeOH (7:3—0:10, *v/v*) to yield four subfractions (POH 11E1–POH 11E4), and compound **7** (t_R_ = 54 min, 3.4 mg) was purified from the POH 11E2 (80 mg) by semipreparative HPLC (25% MeCN). The POH 11F (245 mg) was subjected to chromatography on a C_18_ MPLC using H_2_O and MeOH (6:4—0:10, *v/v*) to yield five subfractions (POH 11F1–POH 11F5), and the POH 11F1 (173 mg) was purified by semipreparative HPLC (20% MeCN) to obtain compounds **2** (t_R_ = 47 min, 5.1 mg) and **4** (t_R_ = 53 min, 68 mg).

The ethyl acetate fraction (150 g) was subjected to column chromatography (CC) on silica gel (40 × 8 cm, 200–400 mesh, Merck, New Jersey, USA) with a chloroform–methanol gradient (each 1L, 1:0—0:1, *v/v*) to yield nine fractions (POE 1–POE 9). POE 2, 3, 4, 5, 6, and 8 were applied to silica gel CC again and eluted with a chloroform–methanol gradient (100:0—0:100, *v/v*). POE 1 was subjected to chromatography on a SiO_2_ MPLC with a chloroform–methanol gradient (100:0—0:100, *v/v*) and subfractions were obtained.

The POE 1E (503 mg) was subjected to chromatography on a SiO_2_ MPLC using H_2_O and MeOH (1:9—0:10, *v/v*) to yield 12 subfractions (POE 1E1–POE 1E12). Compound **20** (t_R_ = 54 min, 1.9 mg) was purified from the POE 1E5 (85 mg) by semipreparative HPLC (35% MeCN). The POE 2A (240 mg) was subjected to chromatography on a C_18_ MPLC using H_2_O and MeOH (2:8—0:10, *v/v*) to obtain nine subfractions (POE 2A1–POE 2A9), and POE 2A2 (42 mg) was purified from semipreparative HPLC (20% MeCN) to isolate compound **17** (t_R_ = 48 min, 2.4 mg). The POE 2B (2.6 g) was chromatographed on a C_18_ MPLC using H_2_O and MeOH (2:8—0:10, *v/v*) to yield 12 subfractions (POE 2B1–POE 2B12). The POE 2B2 (220 mg) was obtained by semipreparative HPLC (15% MeCN) to obtain compound **22** (t_R_ = 57 min, 25 mg). The POE 2C (2.4 g) was subjected to chromatography on a C_18_ MPLC using H_2_O and MeOH (25:75—0:100, *v/v*) to obtain 13 subfractions (POE 2C1–POE 2C13). Compound **9** (t_R_ = 64 min, 8.0 mg) was isolated from POE 2C7 (147 mg) by semipreparative HPLC (69% MeCN). The POE 2D (2.8 g) was chromatographed on a C_18_ MPLC using H_2_O and MeOH (25:75—0:100, *v/v*) to gain 11 subfractions (POE 2D1–POE 2D11). The POE 2D4 (119 mg) was purified by semipreparative HPLC (20% MeCN) to obtain compound **16** (t_R_ = 66 min, 7.7 mg). The POE 2D8 (291 mg) was isolated by semipreparative HPLC (40% MeCN) to obtain compound **10** (t_R_ = 37 min, 11.8 mg). Compounds **12** (t_R_ = 50 min, 4.8 mg), **13** (t_R_ = 58 min, 5.2 mg), and **14** (t_R_ = 70 min, 2.5 mg) were purified from POE 3B (504 mg) by semipreparative HPLC (25% MeCN). The POE 3D (687 mg) was purified by semipreparative HPLC (30% MeCN) to obtain compound **8** (t_R_ = 78 min, 5.6 mg). The POE 3E (2.6 g) was subjected to chromatography on a C_18_ MPLC using H_2_O and MeOH (85:100–0:100, *v/v*) to gain 14 subfractions, and **15** (t_R_ = 65 min, 4.1 mg). Compound **5** (t_R_ = 54 min, 14.5 mg) was obtained from the POE 4D (102 mg) by semipreparative HPLC (25% MeCN). Compound **21** (t_R_ = 41 min, 10.4 mg) was isolated from POE 5A (201 mg) by semipreparative HPLC (15% MeCN).

Compound **8**: White solid; ${\left[\alpha \right]}_{D}^{25}$ = −18 (*c* 0.1, MeOH); HRESIMS *m*/*z* 396.1422 [M + Na]^+^ (396.1423, calcd for C_20_H_23_O_6_Na); ^1^H NMR data in methanol-*d*_4_ (600 MHz) and ^13^C NMR data in methanol-*d*_4_ (150 MHz) are reported in Table 1.

Compound **15**: Brown liquid; ${\left[\alpha \right]}_{D}^{25}$ = + 15 (*c* 0.1, MeOH); ESI-MS *m/z* 273.8 [M-H]^+^; ^1^H NMR (methanol-*d*_4_, 600 MHz) δ_H_ 9.40 (1H, s, H-7), 6.94 (1H, d, *J* = 4.2 Hz, H-3), 6.57 (1H, d, *J* = 7.8 Hz, H-2’), 6.47 (1H, d, *J* = 2.4 Hz, H-5’), 6.34 (1H, dd, *J* = 7.8, 2.4 Hz, H-6’), 6.14 (1H, d, *J* = 4.2 Hz, H-4), 4.35 (2H, t, *J* = 7.8 Hz, H-8’), 4.07 (2H, s, H-6), 3.22 (3H, s, OCH_3_-6), 2.75 (2H, t, *J* = 7.2 Hz, H-7’); ^13^C NMR (methanol-*d*_4_, 150 MHz) δ_C_ 181.1 (C-7), 146.5 (C-3’), 145.2 (C-4’), 141.6 (C-5), 133.8 (C-2), 131.5 (C-1’), 124.5 (C-3), 121.3 (C-6’), 117.2 (C-2’), 116.5 (C-5’), 112.6 (C-4), 66.6 (C-6), 58.3 (OCH_3_-6), 49.8 (C-8’), 38.3 (C-7’), (Appendix A).

Compound **20**: Colorless needles; ${\left[\alpha \right]}_{D}^{25}$ = + 20 (*c* 0.1, MeOH); ESI-MS *m/z* 227.0 [M + H]^+^; ^1^H NMR (methanol-*d*_4_, 600 MHz) δ_H_ 8.45 (1H, d, *J* = 5.4 Hz, H-4), 8.35 (1H, d, *J* = 5.4 Hz, H-3), 8.25 (1H, d, *J* = 7.8 Hz, H-5), 7.74 (1H, d, *J* = 7.8 Hz, H-8), 7.63 (1H, m, H-6), 7.34 (1H, m, H-7), 4.12 (3H, s, OCH_3_-1’); ^13^C NMR (methanol-*d*_4_, 150 MHz) δ_C_ 164.1 (C-1’), 153.8 (C-1), 153.7 (C-3), 138.9 (C-13), 133.7 (C-10), 130.8 (C-12), 130.6 (C-6), 122.9 (C-5), 121.9 (C-11), 121.8 (C-7), 120.1 (C-4), 113.6 (C-8), 53.1 (OCH_3_-1’), (Appendix A).

### 3.4. Cell Culture

RAW 264.7 (ATCC TIB-71) cells were cultured in Dulbecco’s modified Eagle medium (DMEM) and RPMI 1640 medium supplemented with 10% fetal bovine serum, 2 mM glutamine, 100 U/mL penicillin, and 100 mg/mL streptomycin sulfate. Cell were maintained at 37 °C in humidified air with 5% CO_2_.

### 3.5. Measurement of NO Contents and Cell Cytotoxicity

RAW 264.7 (ATCC TIB-71) cells were cultured in Dulbecco’s modified Eagle medium (DMEM) and NO assay was carried out for measurements of NO release using previously reported method [46]. Briefly, RAW 264.7 cells were plated at 1 × 10^5^ cell density in a 96-well microplate and cultured for 24 h. Compounds (**1**–**22**) were pretreated with increasing dose concentrations (0.5, 1, 5, 10, 25, 50, and 100 μM), and then stimulated with LPS (1 μg/mL, Sigma–Aldrich, St. Louis, MO, USA) for 18 h. The mixture of cell supernatant (100 μL) and Griess reagent [1% sulfanilamide + 0.1% *N*-(1-naphthyl)ethylenediamine (Sigma–Aldrich)] in 5% phosphoric acid was recorded at 550 nm using a microplate reader (Varioskan LUX, Thermo Fisher Scientific Inc., Waltham, MA, USA). RAW 264.7 cell cytotoxicity was evaluated using 3-(4,5-dimethylthiazol-2-yl)-2,5-diphenyltetrazolium bromide (MTT) assay [47].

### 3.6. Immunoblot Analysis

The whole cell lysate was extracted using Cell Lysis Buffer (Cell Signaling Technology, Beverly, MA, USA). The cytosolic and nuclear extracts were prepared using an NE-PER Nuclear and Cytoplasmic Extraction Kit (Thermo Fisher Scientific) according to the manufacturer’s protocol. Immunoblot analysis was performed as previously described [48]. After transfer to nitrocellulose (NC) membrane, the blocking membrane with 5% skimmed milk powder was incubated overnight at 4 °C with primary antibody, including anti-phospho-JNK (1:1000), anti-JNK (1:1000), anti-phospho-p38 (1:1000), anti-p38 (1:1000), anti-phospho-ERK (1:1000), anti-ERK (1:1000), anti-p65 (1:1000), anti- IκBα (1:1000), anti-iNOS (1:1000), and anti-β-actin antibodies (Cell Signaling Technology, Beverly, MA, USA). The membranes were then incubated with a horseradish peroxide-conjugated anti-rabbit secondary antibody (1:5000) at room temperature. The band densities were calculated with Quantity One software (Bio-Rad Laboratories, Hercules, CA, USA).

### 3.7. Real-Time PCR Using TaqMan Probe

Total RNA was extracted from RAW 264.7 cells using the TaKaRa MiniBEST Universal RNA Extraction Kit following the manufacturer’s instructions (TaKaRa Bio Inc., Shiga, Japan). The complementary DNA (cDNA) was synthesized from 1 μg of the total RNA using a PrimeScript 1st strand cDNA synthesis kit (Takara Bio Inc. Japan). Quantitative real-time PCR (qPCR) of Il1bβ (Mm00434228_m1), Il6 (Mm00446190_m1), and Tnf (Mm00443258_m1) was performed with a TaqMan Gene Expression Assay Kit (Thermo Fisher Scientific, San Jose, CA, USA). To normalize the gene expression, an 18S rRNA endogenous control (Applied Biosystems, Foster City, CA, USA) was used. The qPCR was employed to verify the mRNA expression using a Step-One Plus Real-Time PCR system. To quantify mRNA expression, TaqMan mRNA assay was performed according to the manufacturer’s protocol (Applied Biosystems) [49]. PCR amplification was analyzed using the comparative ΔΔCT method.

### 3.8. Immunocytochemistry

RAW 264.7 (1 × 10^6^ per well) cells were seeded on collagen-coated coverslips in six-well plates and incubated overnight. Then, the cells were pretreated with compound **20** (12.5 µM) or dexamethasone (10 µM) for 2 h. Then, the coverslips were washed with phosphate-buffered saline (PBS) and fixed with ice-cold methanol for 10 min at room temperature. After blocking with 1% bovine serum albumin (BSA) in PBS containing 0.1% Tween 20 (PBST) for 30 min, the coverslips were incubated overnight with an anti-p65 NF-κB antibody in a humidified chamber at 4 °C, followed by coincubation with an FITC-labeled anti-rabbit IgG antibody in a 37 °C humidified chamber for 60 min in the dark. The coverslips were sealed and stained with DAPI by using ProLong Gold antifade reagent with DAPI (Thermo Fisher Scientific, Waltham, MA, USA). Immunofluorescence images were obtained using an Olympus IX73 microscope with cellSens software (Olympus, Center Valley, PA, USA) [47].

### 3.9. Statistical Analysis

GraphPad Prism 5 software (GraphPad software, San Diego, CA, USA) was used for the statistical analysis of the experimental results. Each experiment, including NO assay, MTT assay, immunoblot, and real-time PCR, was performed independently three times, and these data represent the mean ± standard error of the mean (SEM). For comparisons between the control and LPS-treated groups, the unpaired *t*-test or Mann–Whitney *U*-test was used according to the data distribution. The statistical significance of each value was measured by the unpaired Student’s *t*-test or Mann–Whitney *U-*test. * *p* < 0.05, ** *p* < 0.01, and *** *p* < 0.001 were considered significant.

## 4. Conclusions

The 1-carbomethoxy-β-carboline (compound **20**), which has an alkaloid structure as an active compound responsible for the anti-inflammatory activity of *P. oleracea* extract (95%), disturbed one of the major intracellular inflammatory signaling pathways associated with MAPKs and suppressed the nuclear translocation of NF-κB, decreasing proinflammatory mediators such as iNOS, TNF-α, IL-6, and IL-1β. These results indicate that *P. oleracea* extract and its components, specifically alkaloid compound **20**, may be useful and safe therapeutic agents for treating the painful symptoms of inflammatory diseases such as rheumatoid arthritis, allergic asthma, and atopic dermatitis as an alternative to NSAIDs and dexamethasone. Further studies are necessary to improve the extraction efficiency of compound **20** and determine the structural characteristics responsible for its bioactivity and to evaluate better clinical mimic experiments such as using human monocytes rather than the single mouse cell line, RAW 264.7.

## Figures and Tables

**Figure 1 molecules-24-04042-f001:**
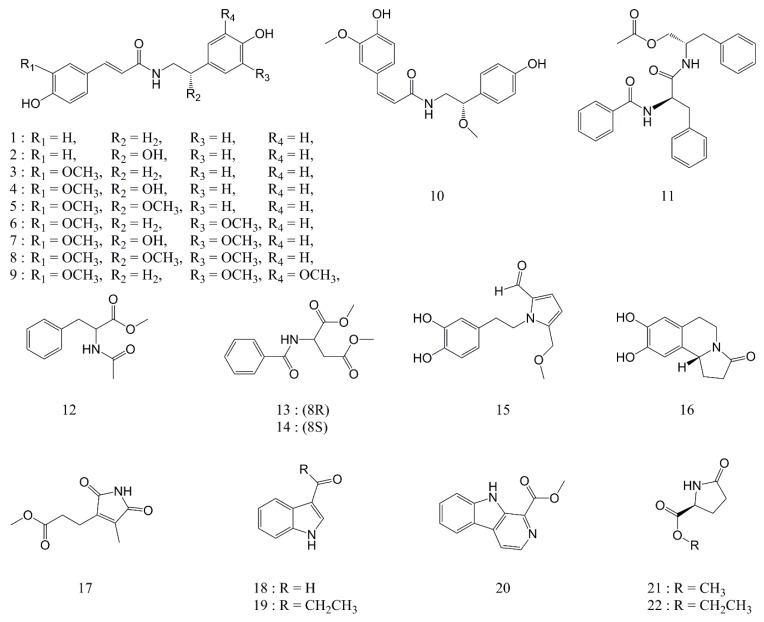
Chemical structures of the isolated compounds (**1**–**22**).

**Figure 2 molecules-24-04042-f002:**
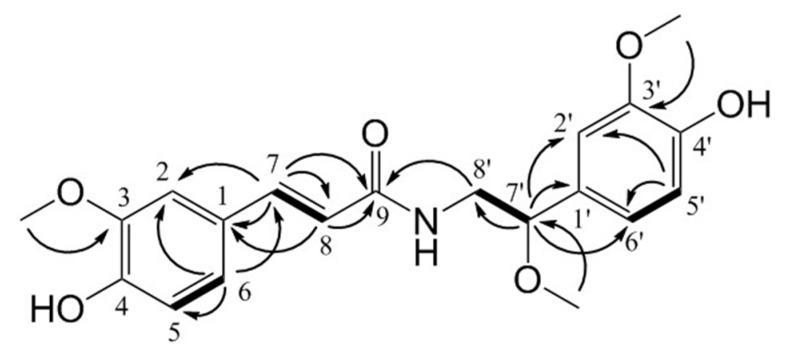
Selected key HMBC (plain arrow) and ^1^H-^1^H COSY (bold line) correlations for **8**.

**Figure 3 molecules-24-04042-f003:**
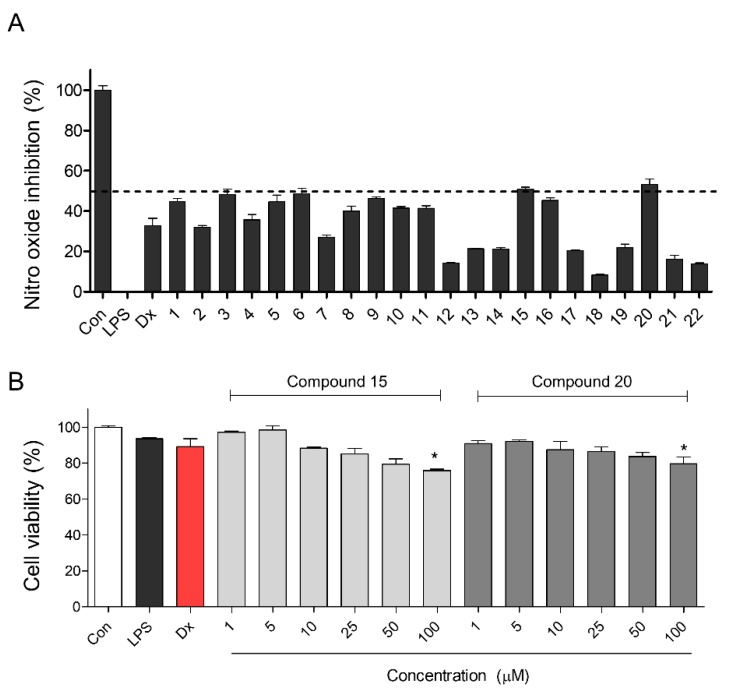
Nitric oxide screening in *P. oleracea* constituents in RAW 264.7 cell line. (**A**) NO screening was evaluated among 22 constituents in *P. oleracea.* NO screening was performed as triplicate tests, and results are expressed as means ± standard error of the mean (SEM). (**B**) For selected compounds 15 and 20, MTT assay was performed at 1 to 100 μM. MTT assay was performed as triplicate tests, and results are expressed as means ± SEM. An unpaired Student’s *t-*test was used for statistical analysis. * *p* < 0.05 versus Con. Con: control, LPS: lipopolysaccharide, Dx: dexamethasone.

**Figure 4 molecules-24-04042-f004:**
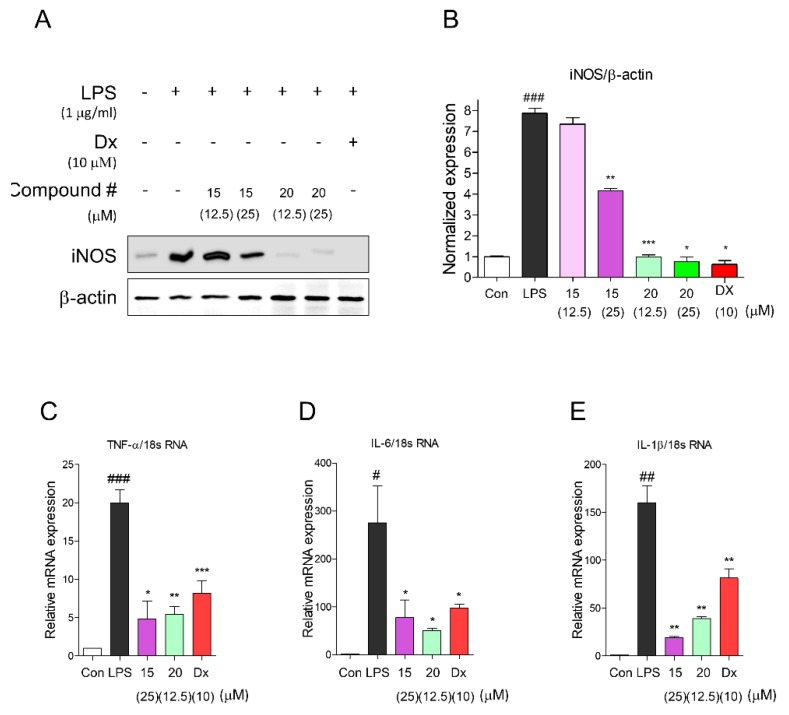
Compound **15** and **20** showed anti-inflammatory effects by inhibiting proinflammatory mediators. (**A**) Compounds **15** and **20** decreased iNOS expression levels in LPS-induced RAW 264.7 cells. (**B**) Relative ratio of iNOS versus β-actin was measured using densitometry, and dexamethasone was used as positive control. These graphs represented that compounds **15** and **20** dose-dependently inhibited iNOS using immunoblot analysis. Cells were pretreated with each compound for 2 h and stimulated with LPS (1 μg/mL) for 16 h. Immunoblot analysis was performed in triplicate tests, and results are expressed as means ± SEM. An unpaired Student’s *t-*test was used for statistical analysis. ### *p* < 0.001 versus Con, * *p* < 0.05, ** *p* < 0.01, and *** *p* < 0.001 versus LPS. (**C**–**E**) The mRNA expression levels of TNF-α, IL-6, and IL-1β were measured using quantitative real-time PCR experiment, and these proinflammatory cytokines were significantly diminished by compounds **15** and **21**. Cells were preincubated for 2 h with compounds **1** and **3** at concentrations of 5 and 10 μM, respectively, and activated by LPS (1 μg/mL) for 2 h. Results represented as mean ± SEM, and dexamethasone was used as a positive control. # *p* < 0.05, ## *p* < 0.01, ### *p* < 0.001 versus Con, * *p* < 0.05, ** *p* < 0.01, and *** *p* < 0.001 versus LPS. Con: control, LPS: lipopolysaccharide, Dx: dexamethasone.

**Figure 5 molecules-24-04042-f005:**
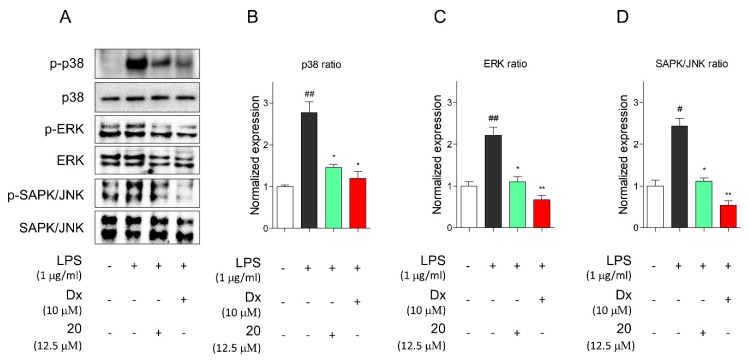
Compound **20** suppressed MAPK signaling pathway. (**A**) Immunoblot analysis showed that phosphorylated protein levels of MAPKs pathway, p38, ERK, and JNK were inhibited by compounds **20** in RAW 264.7 macrophages. (**B**–**D**) The graphs represent ratio of protein level of p38 (**B**), ERK (**C**), and JNK (**D**). Cells were preincubated for 2 h with compound **20** at concentration of 12.5 μM and stimulated with LPS (1 μg/mL) for 1 h. Dexamethasone served as positive control. Immunoblot analysis performed as triplicate experiments, and data represented as means ± SEM. Significant difference was considered at the levels of # *p* < 0.05 and ## *p* < 0.01 versus Con, * *p* < 0.05 and ** *p* < 0.01 versus LPS. Con: control, LPS: lipopolysaccharide, Dx: dexamethasone.

**Figure 6 molecules-24-04042-f006:**
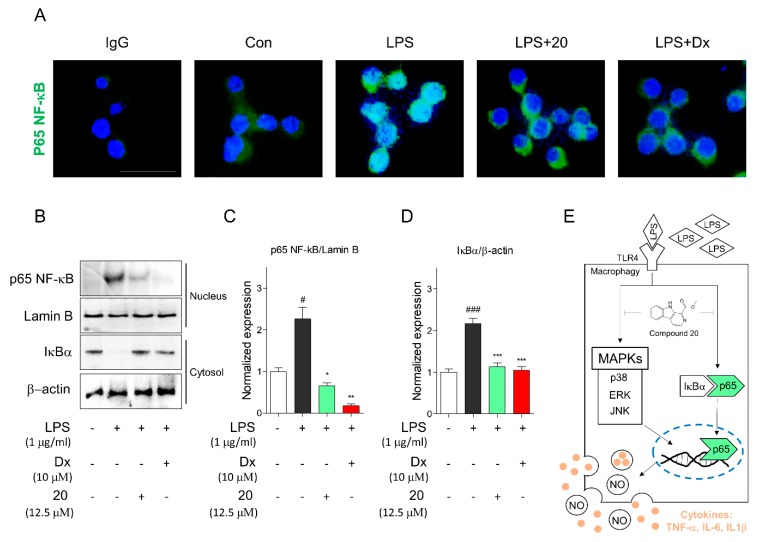
Compound **20** suppressed activation of NF-κB. (**A**) Compound **20** suppressed nuclear translocation of NF-κB. After 2 h of compound **20** treatment, the cells were fixed and permeabilized. NF-κB (green) was immunostained with rabbit-anti-NF-κB followed by FITC-conjugated secondary antibodies and the nuclei (blue) were stained with DAPI. The results shown are representative of two independent experiments. (**B**) Immunoblot analysis displayed that translocation of NF-κB into nucleus and degradation of cytosolic IκBα were suppressed by compound **20** in RAW 264.7 cells. (**C**) The graph is described as relative ratio of NF-κB to lamin B. (**D**) The graph shows the relative ratio of IκBα to β-actin using densitometry of ImageJ. (**E**) The schematic pathway of compound **20** is exhibited. Cells were pretreated for 2 h with compound **20** at concentration of 12.5 μM, and stimulated with LPS (1 μg/mL) for 1 h. Dexamethasone was used as positive control, and immunoblot analysis was performed as triplicate experiments. Values are means ± SEM, and an unpaired Student’s *t-*test was used for statistical analysis. # *p* < 0.05, and ### *p* < 0.001 versus Con, * *p* < 0.05, ** *p* < 0.01, and *** *p* < 0.001 represented significant differences from the LPS-treated group. Con: control, LPS: lipopolysaccharide, Dx: dexamethasone.

**Table 1 molecules-24-04042-t001:** Spectroscopic data for compound **8**
^a^ in methanol-*d*_4_.

Positions	δ_C_ ^c^	Type	δ_H_ (*J* in Hz) ^b^
1	127.0	C	
2	110.2	CH	7.13 (1H, d, 1.8)
3	148.8	C	
4	148.5	C	
5	115.1	CH	6.80 (1H, overlap)
6	121.9	CH	7.03 (1H, dd, 8.4, 1.8)
7	140.9	CH	7.44 (1H, d, 15.6)
8	117.3	CH	6.45 (1H, d, 15.6)
9	167.9	C	
OCH_3_-3	55.6	CH_3_	3.89 (3H, s)
1’	130.9	C	
2’	109.8	CH_3_	6.92 (3H, s)
3’	147.9	C	
4’	146.2	C	
5’	114.8	CH	6.80 (1H, overlap)
6’	119.5	CH	6.80 (1H, overlap)
7’	82.2	CH	4.26 (1H, dd, 8.4, 4.8)
8’α	48.2	CH_2_	3.53 (1H, dd, 13.8, 4.8)
8’β			3.42 (1H, dd, 13.8, 8.4)
OCH_3_-3’	55.1	CH_3_	3.86 (3H, s)
OCH_3_-7’	45.7	CH_3_	3.24 (3H, s)

^a^ TMS was used as an internal standard; chemical shifts (δ) are reported in ppm; *J* values are reported in Hz. Data were measured in methanol–*d*_4_ at ^b^ 600 MHz or ^c^ 150 MHz.

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
