# Peer review of "1-Carbomethoxy-β-Carboline, Derived from Portulaca oleracea L., Ameliorates LPS-Mediated Inflammatory Response Associated with MAPK Signaling and Nuclear Translocation of NF-κB"

_molecules, 2019, doi:10.3390/molecules24224042_

Round 1
Reviewer 1 Report
In this work, authors elucidated the chemical structures of the phytochemical substituents using NMR and ESI-MS and screened their effects on NO production in LPS-induced macrophages. The work is very interesting. Authors identified a new compound in this species and this is the first study describing the capacity of 1-carbomethoxy-β-carboline to ameliorate the NO production and pro-inflammatory cytokines by inhibiting the MAPK pathway and the nuclear translocation of NF-κB under LPS mediated inflammatory conditions in RAW 264.7. Therefore, only minor concerns about the work are described below.
Authors should provide the authority for each species.
Authors concluded that "(...) the major constituents of P. oleracea extract responsible for its ability to inhibit NO production were compounds 15 and 20". Nonetheless, compounds 3 and 6 seem to have a very similar activity than compounds 15 and 20. Authors should explain better this selection.
Concerning statistical analysis, information about the normality of distribution and homogeneity of variance of the data should be provided.
Conclusion should be more concise.
Author Response
Reviewer 1
Authors concluded that "(...) the major constituents of P. oleracea extract responsible for its ability to inhibit NO production were compounds 15 and 20". Nonetheless, compounds 3 and 6 seem to have a very similar activity than compounds 15 and 20. Authors should explain better this selection.
: We additionally explained previous studies about compounds 3, 5, 6, 9, and 16 as a reasons to decide compound 15 and 20. Please refer in line: 151 to 158 into the manuscript of ‘Result and Discussion’.
“In NO production screening in Figure. 3, Compound 3, 5, 6, 9, and 16 showed similar inhibitory effect of NO production such as compound 15 and 20. However, compound 3, 5, 6, 9, and 16 were already reported previous studies associated with anti-inflammatory effects or cellular toxicity. Compound 3, derived from Tinospora crispa and Wolfberry and compound 6, derived from Wolfberry previously reported to inhibit NO production [1, 2]. Compound 5, derived from Solanum nigrum, had anti-inflammatory effect by inhibition of leukotriene C4 (LTC4) [3] and compound 16, derived from P. oleracea had neuroprotective effect by reducing reactive oxygen species (ROS) and inhibiting ERK 1/2 phosphorlyation [4]. Compound 9, derived from Hibiscus tiliaceus, had cytotoxic effect [5].”
Concerning statistical analysis, information about the normality of distribution and homogeneity of variance of the data should be provided.
: We added statistical explanation of analysis. Please refer in line: 390 to 393 into the manuscript of ‘Materials and Methods’.
“For comparisons between the control and LPS-treated groups, the unpaired t test or Mann-Whitney U test was used according to the data distribution. The statistical significance of each value was measured by the unpaired Student`s t test t test or Mann-Whitney U test. *p < 0.05, **p < 0.01, and ***p < 0.001 were considered significant.”
Conclusion should be more concise.
: Our group revised manuscript of ‘Conclusion’ to focus on the effects of compound 20 and necessity of development for anti-inflammatory therapeutic agent. Therefore, we deleted further explanation of future potential for compound 20 and of overlapped sentences in line: 403 to 409.

Reviewer 2 Report
The authors have identified the components of ethanol-extract from P. oleracea. Compounds have been identified and screened for macrophage inhibition. The study provides basic understanding of new therapeutics that can be eventually studied for its therapeutic activity. The study is within the scope of the journal.
The study, overall, feels very limited in its biological studies. Following suggestions may improve its significance and its interest to the readers.
Compounds were selected based on a 50% inhibition of NO production by macrophages. A description on how this 50% cut-off value was decided needs to be given. Furthermore, compounds, 3,5,6,9 and 16, also shows comparable inhibition (close to 50% inhibition) as compounds 15 and 20. Were they further tested as well? What do the authors mean by "Con" in Figures 3 and 4? Were these RAW cells cultured just in growth media? All the immunological studies provided are restricted to a single mouse cell line. Tests should also be done in primary immune cells as well as in human monocytes to understand the drug's relevance to clinics. Was the compound 20 used in all the assays purified from the plant extract? How does its anti-macrophage activity compare to synthetically prepared 1-carbomethoxy-β-carboline? The effect of compound 20 on polarization of macrophages to M1 to M2 phenotype should be tested. Is the compound 15 or 20 effective in reducing macrophage activation during inflammation in an experimental animal model, such as collagen-induced arthritis in mice?
Author Response
Reviewer 2
1) Compounds were selected based on a 50% inhibition of NO production by macrophages. A description on how this 50% cut-off value was decided needs to be given. Furthermore, compounds, 3,5,6,9 and 16, also shows comparable inhibition (close to 50% inhibition) as compounds 15 and 20. Were they further tested as well?
: Among major nonsteroidal anti-inflammatory drugs such as aspirin, fenoprofen, indomethacine, etodolac, piroxicam, aceclofenac, diclofenac, sulindac, and phenylbutazone, phenylbutazone of 100 μg/ml was best inhibitory effect of nitric oxide about 50% in RAW 264.7 cells [6]. That was why we decided 50% cut-off value. We explained this information in line: Second, the academic information of compounds 3, 5, 6, 9, and 16 equally was asked by reviewer 1. Please refer to comment 1 of reviewer 1. Please refer in line: 145 to 147 into the manuscript of ‘Result and Discussion’.
“Because, phenylbutazone, one of the nonsteroidal anti-inflammtory drugs, of 100 μg/ml was best inhibitory effect of nitric oxide about 50% in RAW 264.7 cells [6].”
2) What do the authors mean by "Con" in Figures 3 and 4? Were these RAW cells cultured just in growth media?
: As we described in figure legend, “Con” is control as a vehicle, which cultured just in growth media.
3) All the immunological studies provided are restricted to a single mouse cell line. Tests should also be done in primary immune cells as well as in human monocytes to understand the drug's relevance to clinics.
: We recognized academic evaluation for clinical mimics such as primary immune cells or human monocytes. However, we did not set-up to evaluate experimental clinical mimics yet. Because our institute should primarily approve for clinical experiments from institutional review board (IRB). Thus, we added context of experimental limitation using single mouse cell line, RAW 264.7, and discussed further study for clinical evaluation on development of drugs from natural constituents in ‘Conclusion’. Please refer line: 404 into the manuscript of ‘Conclusion.
“to evaluate better clinical mimic experiments such as human monocytes than single mouse cell line, RAW 264.7”
4) Was the compound 20 used in all the assays purified from the plant extract? How does its anti-macrophage activity compare to synthetically prepared 1- carbomethoxy- β-carboline?
: As reviewer 2 mentioned, we considered consistent effects of anti-inflammation between natural compound of 1-carbomethoxy-β-carboline and synthetic compound of 1-carbomethoxy-β-carboline. Thus, our group only focused on anti-inflammatory effects about natural constituents of P. oleracea extract and we consistently tested all using directly purified compound 20 from P. oleracea extract.
5) The effect of compound 20 on polarization of macrophages to M1 to M2 phenotype should be tested.
: The polarization states of macrophages are largely categorized as classically activated M1 macrophages or alternatively activated M2 macrophages [7]. According to results of Figure 3, the treatment of compound 20 (50 μM) highly inhibited NO production and there were no cytotoxicity under 50 μM. M2 macrophages are known to inhibit inflammation and promote tissue remodeling and angiogenesis [8]. Our group guessed that compound 20 made macrophages, RAW 264.7 cells, to polarize M1 to M2 by inhibition of NO production, iNOS expression, and pro-inflammatory cytokines. In this study, we highly emphasized to perform separate purification of P. oleracea extracts and to evaluate active compound of anti-inflammatory action in P. oleracea extracts rather than macrophage phenotype.
6) Is the compound 15 or 20 effective in reducing macrophage activation during inflammation in an experimental animal model, such as collagen- induced arthritis in mice?
: Because compound 20 decreased pro-inflammatory mediators, we predicted available therapeutic agent for inflammatory diseases associated with MAPKs and NF-κB. However, we did not evaluated in an experimental animal model. To clear misunderstanding, we deleted context of the prediction of non-tested effects for specific disease, line: 206.
Reviewer 3 Report
The manuscript entitled “1-Carbomethoxy-β-carboline, derived from Portulaca oleracea L., ameliorates LPS-mediated inflammation by suppressing MAPKs signaling and nuclear translocation of NF-κB” by Kang-Hoon Kim, Eun-Jae Park, Hyun-Jae Jang, Seung-Jae Lee, Chan Sun Park, Bong-Sik Yun, Seung Woong Lee, and Mun-Chual Rho discusses the impact of P. oleracea compounds on NF-κB signaling and proinflammatory response associated with MAPK and NF-κB. This study designates 1-Carbomethoxy-β-carboline as a promising antiinflammatory agent.
The results are interesting. However, this paper needs to be improved.
(1) The main concern is the title of the manuscript and the conclusions, which, in my opinion, go too far. The Authors did not show the direct link between the MAPK and NF-κB signaling and the response such as cytokine expression. NF-κB is not the sole regulator of the proinflammatory cytokine expression. For example, LPS-induced IL-1β expression is also regulated by AP-1 transcription factor. Therefore, the Authors selected only certain components of the inflammatory response activation which are related to NF-κB. The other pathways may also be involved. I would recommend changing the title, for example: "1-Carbomethoxy-β-carboline, derived from Portulaca oleracea L., ameliorates LPS-mediated inflammatory response associated with MAPKs signaling and nuclear translocation of NF-κB". Due to the same reason, the lines: 25, 81, 163, 164, 192, 194, 220, 386 need to be rewritten.
(2) Fig. 6: The Authors do not show p65 cytoplasmic content on western blots. It would be, however, sufficient to add an appropriate IF image of negative Con (untreated cells).
(3) Other minor improvements needed:
Line 56: kappa --> κ
Line 62: toll-like --> Toll-like
Lines 155, 179, 209, 232, 328, 333, 335, 341, 354: RAW264.7 --> RAW 264.7
Line 213: delete ##p<0.01
Line 225 (Fig. 6E): Macrophagy ---> Macrophage
Line 226: Compounds 20 suppressed nuclear translocation of NF-κB --> Compound 20 suppressed activation of NF-κB
Lines 328, 333: was --> were
Line 337: than --> then
Line 345: Please add the information about obtaining nuclear and cytoplasmic fractions
Line 349: Cell Signaling --> Cell Signaling Technology
Line 352: delete "."
Line 365: Provide cell number
Lines 366, 367: please correct information about cell treatment.
Overall, this manuscript can be considered for publication after introducing necessary changes.
Author Response
Reviewer 3
1) The main concern is the title of the manuscript and the conclusions, which, in my opinion, go too far. The Authors did not show the direct link between the MAPK and NF-κB signaling and the response such as cytokine expression. NF- κB is not the sole regulator of the proinflammatory cytokine expression. For example, LPS-induced IL-1β expression is also regulated by AP-1 transcription factor. Therefore, the Authors selected only certain components of the inflammatory response activation which are related to NF-κB. The other pathways may also be involved. I would recommend changing the title, for example: "1-Carbomethoxy-β-carboline, derived from Portulaca oleracea L., ameliorates LPS-mediated inflammatory response associated with MAPKs signaling and nuclear translocation of NF-κB". Due to the same reason, the lines: 25, 81, 163, 164, 192, 194, 220, 386 need to be rewritten.
: 1-1) As Reviewer 3 mentioned, we revised title as “1-Carbomethoxy-β-carboline, derived from Portulaca oleracea L., ameliorates LPS-mediated inflammatory response associated with MAPKs signaling and nuclear translocation of NF-κB”.
: 1-2) Although AP-1 transcription factor regulated pro-inflammatory cytokines under LPS-induced condition in macrophage, canonical NF-κB pathway had been defined primarily in response to TNF-a and IL-1 signaling, prototypical pro-inflammatory cytokines on pathogenesis of chronic inflammatory diseases such as rheumatoid arthritis, inflammatory bowel disease, asthma, and chronic obstructive pulmonary disease [9-11]. That’s why we tested for certain factor as a nuclear translocation of NF-κB. However, as reviewer 2 advised, other pathways may also be possibly involved. Thus, we euphemistically revised the sentences of 25, 81, 172, 173, 203, 204, 229, and 397.
2) Fig. 6: The Authors do not show p65 cytoplasmic content on western blots. It would be, however, sufficient to add an appropriate IF image of negative Con (untreated cells).
: We replaced high resolution image of IF image of negative control in Figure 6.
3) Other minor improvements needed
: As you kindly mentions, we exactly revised all minor errors of sentences.
Round 2
Reviewer 3 Report
The manuscrpit needs minor improvements:
Line 202: delete "inflammatory"
Line 233 (Fig. 6E): macrophagy --> macrophage